# Reproducibility Study - SCOUTER: Slot Attention-based Classifier for Explainable Image Recognition

## Reproducibility Summary

**Scope of Reproducibility**

We aim to replicate the main findings of the paper *SCOUTER: Slot Attention-based Classifier for Explainable Image Recognition* by Li et al. in order to verify the main claims they make: 1) The explanations generated by SCOUTER outperform those by other explanation methods in several explanation evaluation metrics. 2) SCOUTER achieves similar classification accuracy as a fully connected model. 3) SCOUTER achieves higher confusion matrix metrics than a fully connected model on a binary classification problem.

**Methodology**

The authors provided code for training the models. We implemented the explanation evaluation metrics and confusion matrix metrics ourselves. We used the same hyperparameters as the original work, in case the hyperparameter was reported. We trained all models from scratch on various datasets and evaluated the explanations generated by these models with all reported metrics. We compared the accuracy scores between different models on several datasets. Finally, we calculated an assortment of confusion matrix metrics on models trained on a binary dataset.

**Results**

We were only able to reproduce 22.2% of the explanation evaluation metrics and could thus not find conclusive support for claim 1. We could only verify claim 2 for one of the datasets and in total could reproduce 55.5% of the original scores. We could reproduce all scores regarding claim 3, but the claim is still not justified, as the scores between the fully connected and SCOUTER models lie very close to one another.

**What was easy**

The paper was well written, so understanding the SCOUTER architecture was straightforward. The code for training a model was available and together with the examples the authors provide, this was achievable with relative ease. A checkpoint system is implemented, so training a model can be split into multiple runs. All used datasets are available and straightforward to obtain.

**What was difficult**

The original code did not contain any documentation, which made it difficult to navigate. No code for calculating the metrics was provided and this had to be implemented from scratch. During the training of the models, memory allocation issues occurred. Training and evaluating on a large dataset took a considerable amount of time.

**Communication with original authors**

We sent the authors an e-mail to request either the missing code or more details on how the metrics were implemented, but unfortunately we did not receive a reply.

# 1 Introduction

Explainable Artificial Intelligence (XAI) is growing in popularity and becomes increasingly important as more and more AI applications are used in daily life. It is important to visualize both positive and negative patterns in the explanation of a model [2], but this discernment has not gained much attention yet. In [8], Li et al. introduce a model architecture that is capable of generating both positive and negative explanations based on an explainable slot attention module.

# 2 Scope of reproducibility

The authors sought to tackle the problem of deep neural networks being unintelligible. For this purpose they developed SCOUTER (Slot-based COnfigUrable and Transparent classifiER) [8]. The unique aspect of SCOUTER is that every category has its corresponding positive or negative explanation as to why a particular image does or does not belong to a certain category. This offers a more in-depth look into what a model bases its predictions on and thus increases its explainability.

The main claim of the original paper is aptly summarised in the last sentence of its conclusion: *"Experimental results prove that SCOUTER can give accurate explanations while keeping good classification performance"*. This is what we will be trying to reproduce. While this claim in itself is vague, the authors compare the score SCOUTER achieves on certain datasets to other methods such as GradCAM [6], RISE [11], I-GOS [12] and IBA [13] and show that SCOUTER achieves a similar or higher score in most metrics. Furthermore, they also train a model where the slot attention is replaced with a fully connected layer as an (unexplainable) baseline to compare SCOUTER to. This can be dissected into the three following claims that we will attempt to verify by reproducing the experiments of the authors:

1. SCOUTER will achieve the highest score on the following explanation evaluation metrics: area size, precision, insertion area under curve, deletion area under curve, infidelity and sensitivity on the ImageNet dataset [3] compared to other explanation methods.

2. SCOUTER will achieve similar classification accuracy as the FC model trained and validated on the ImageNet, Con-text [7], and CUB-200-2011 [14] datasets

3. SCOUTER will achieve higher ROC-AUC, Accuracy, Precision, Recall, F1-Score and Cohen's Kappa scores than the FC model on the ACRIMA dataset [4].

# 3 Methodology

The original paper provides a link to the Github repository[1] with the code and instructions necessary to train the models which were reported in the paper. However, the code used to evaluate the explanations of the trained models was not included. We therefore had to implement these ourselves. The area size metric was partially implemented in the authors code, where the area size for a single image was calculated. We extended this code to calculate the average area size over the entire validation set. We implemented the following explanation evaluation metrics from various papers ourselves: precision [8], Insertion Area Under Curve (IAUC) [11], Deletion Area Under Curve (DAUC) [11], infidelity [16] and sensitivity [16]. Interesting to note is that the precision metric is defined by the authors themselves. The papers the authors referenced for IAUC and DAUC[2], and infidelity and sensitivity[3] provided code for the metrics implementation. We used these and adapted them slightly to integrate it with the code for SCOUTER to deviate as little as possible from the original experiments.

Furthermore, there was no code available for working with the ACRIMA dataset, so we implemented this ourselves as well.

Using the code of the authors composited with our own code, we conducted our experiments on the GPU nodes of the LISA cluster on SurfSara[4] which uses an Nvidia GeForce 1080Ti, 11GB GDDR5X GPU.

---

[1]https://github.com/wbw520/scouter
[2]https://github.com/eclique/RISE
[3]https://github.com/chihkuanyeh/saliency_evaluation
[4]https://userinfo.surfsara.nl/systems/lisa/description

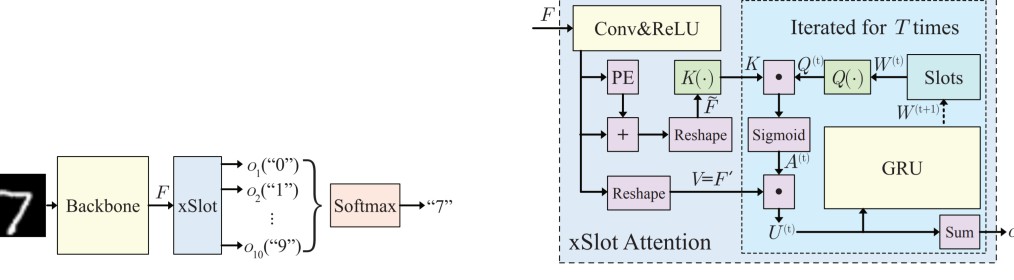

(a) Overview of the classification model.  (b) The xSlot attention module in SCOUTER.

Figure 1: Overview of the SCOUTER model. Taken from Figure 2 in [8].

## 3.1 Model description

Typically a classification model consists of the following: feature extraction using a backbone network, which is then mapped onto a score vector representing the confidence for each class. When using fully connected layers to map such a feature onto a score vector it results in a model that is a black-box, which does not give much information about how or why a certain class attains a higher confidence score.

Such a fully connected classifier is replaced instead by an explainable xSlot module, which is based on an object-centric slot attention module [9]. This creates the SCOUTER model as seen in Figure 1 [8].

In order to reproduce the experiments we will train a multitude of such SCOUTER models. All models use ResNeSt-26 [17] as their backbone, since that was also used in the experiments we aim to reproduce. All models use the same SCOUTER loss as defined by the original authors. Since there are no pre-trained models made available we will train all models from scratch. For all models the amount of parameters is just above 15,000,000. The full table for parameter counts can be found in Table 1.

| Dataset | Fully Connected | SCOUTER |
|---|---|---|
| ImageNet | 15,225,348 | 15,199,584 |
| Con-text | 15,081,918 | 15,195,104 |
| CUB-200-2011 | 15,081,918 | 15,199,584 |
| ACRIMA | 15,024,546 | 15,193,312 |

Table 1: The number of parameters for various models.

## 3.2 Datasets

We used various datasets during our experiments. For reproducing the original experiments we used the ImageNet, Con-text, CUB-200-2011 and ACRIMA datasets. In line with the original experiments, we use the train part of the dataset for the training and the validation part for calculating the metrics.

**ImageNet.** The ImageNet Large Scale Visual Recognition Challenge (ILSVRC) dataset[5] [3] is widely used for classification models. The categories consist of and are organized according to nouns in the WordNet hierarchy [5]. It contains 1,000 categories, 1,281,167 images for training, 50,000 images for validation and 100,000 images for testing. We preprocessed the structure of the directories of the validation set to be in line with the author's code.

**Con-text.** The Con-text dataset[6] [7] is focused on the use of fine-grained classification of buildings into their sub-classes such as cafe, tavern, diner, etc. by detecting scene text in images. The dataset consists of 28 categories with 24,255 images in total. Splitting the dataset was done by the authors using a seed, as there is no inherent split.

**CUB-200-2011.** The Caltech-UCSD Birds 200-2011 (CUB-200-2011) dataset[7] [14] consists of images with photos of 200 bird species (mostly North American). It consists of 200 categories with 11,788 images in total. The train set contains 5,994 images and the test set contains 5,794 images. Note that while the original authors cite the CUB-200

---

[5]Download link: `https://image-net.org/download.php`

[6]Download link: `https://staff.fnwi.uva.nl/s.karaoglu/datasetWeb/Dataset.html`

[7]Download link: `http://www.vision.caltech.edu/visipedia/CUB-200-2011.html`

dataset [15], everything in the available code points towards the authors using the CUB-200-2011 dataset. For example: the code to load the "CUB-200" data is only functional when using the CUB-200-2011 dataset. As such, we made the decision to use CUB-200-2011 for our experiments.

**ACRIMA.** The ACRIMA dataset[8] [4] can be used for automatic glaucoma assessment using fundus images. It contains 2 categories and 705 images. It is composed of 396 glaucomatous images and 309 normal images. There is no inherent split for the data, so we made our own with 80% of the images in the train set and 20% in the validation set using a seed.

### 3.3 Hyperparameters

Many of the hyper-parameters were set in accordance with the original paper, as these were documented and reported. However, not all hyperparameter settings were documented. There is a specific lack of the "slots per class" hyperparameter. We tested both the positive and negative SCOUTER model with four different slots per class hyperparameter settings, namely: 1, 2, 3, and 5. We tested this with $\lambda$ values of both 1 and 10. We found there to be no significant difference in performance in classification accuracy or evaluation metrics between the different slots per class values. As such, all the models that we report on were trained with 1 slot per class since this value was set by the authors in the examples they provided with their code. The full hyperparameter settings can found in Table 2.

| Hyperparameter | Value | Hyperparameter | Value |
|---|---|---|---|
| Epochs | 20 | Lambda Value | {1, 3, 10} |
| Batch Size | 70 | Slots per Class | 1 |
| Number of Classes | $\min(N_{classes}, 100)$ | Power of Slot Loss | 2 |
| Learning Rate | 0.0001 | Image Size | 260 |
| Learning Rate Drop | 70 | Channel | 2048 |
| Hidden Dimensions | 64 | Number of Freeze Layers | 0 |
| Hidden Layers | 3 | Number of Workers | 4 |
| Weight Decay | 0.0001 | World Size | 1 |

Table 2: Hyperparameter settings used for the experiments.

### 3.4 Experimental setup and code

Our code is available at: `https://anonymous.4open.science/r/reproduce_SCOUTER-030C`.

We trained the models using the hyperparameter setup as described above. In order to reproduce the original results we trained six different models for the explanation evaluation: both SCOUTER$_+$ and SCOUTER$_-$ models with $\lambda$ values of 1, 3 and 10.

Wherever classification accuracy is reported it is the accuracy of the model on the validation set after the final epoch, as was done by the original authors. For the evaluation of classification on ImageNet we reused the positive and negative SCOUTER models with a $\lambda$ value of 10, since this model has the same hyperparameter settings. We trained separate positive and negative SCOUTER models with $\lambda = 10$ for the Con-text and CUB-200-2011 classification evaluation. For all datasets we also trained a fully connected classifier model (with ResNeSt-26 as backbone) to compare the SCOUTER models to.

To reproduce the confusion matrix metrics for ACRIMA we trained a positive and negative SCOUTER model on that dataset with $\lambda = 10$.

Results relating to Con-text, CUB-200-2011 and ACRIMA were obtained by averaging the scores from three independent runs. Due to restricted GPU hours and time constraints, we only trained a single model for each configuration on ImageNet.

#### 3.4.1 Metrics

We used several metrics to evaluate the generated explanations. The following metrics were calculated on the ground truth class for SCOUTER$_+$ and on the least similar class for SCOUTER$_-$. The least similar class was determined via Wu-Palmer similarity of the WordNet synsets of the categories as implemented in NLTK [1]. This follows the same formula the original authors used to measure similarity.

**Area size** measures the average size of the generated explanations. This is calculated by summing all the pixel values in

---

[8]Download link: `https://figshare.com/s/c2d31f850af14c5b5232`

the attention map.

**Precision** measures the relative amount of pixels of the attention map that falls within the image's bounding box. Some images in the ImageNet dataset have multiple bounding boxes. We chose to calculate the precision as the max value of each bounding box in the image.

**IAUC** measures the increase in accuracy under the gradual addition of pixels based on their importance in the explanation. The starting state was the image after applying a Gaussian filter of size 11 and $\sigma = 5$.

**DAUC** measures the decrease in accuracy under the gradual removal of pixels based on their importance in the explanation. The final state was an image consisting of only zeroes.

**Infidelity** measures how well the explanation captures the change in the model's prediction under input perturbations. The image was perturbed by adding noise sampled from a unit Gaussian. This metric was calculated over the first 50 images in the validation set.

**Sensitivity** measures how much the explanation is affected by input perturbations. We calculated the maximum sensitivity, as was done in [16]. The image was perturbed by adding noise sampled from a uniform distribution ranging from -0.2 to 0.2. This metric was calculated over the first 50 images in the validation set.

The authors do not give a complete description of how they implemented these metrics. We thus tried to stay as close as possible to the implementations in [11] and [16]. All parameters were thus chosen in accordance with these implementations.

Classification performance was mostly measured via accuracy. The performance of the models on the ACRIMA dataset was evaluated more extensively with several confusion matrix metrics: ROC-AUC, accuracy, precision, recall, F1-score and Cohen's Kappa as implemented in Scikit-learn [10].

### 3.5 Computational requirements

All experiments were conducted on the the GPU nodes of the LISA cluster on SurfSara using a Nvidia GeForce 1080Ti, 11GB GDDR5X.

Computation time varied greatly between datasets. Training a model on the ImageNet dataset took up to 12 hours, for CUB-200-2011 it took around 2 hours, for Con-text it was 1.5 hours and training on the ACRIMA dataset took less than 5 minutes. The calculation of the explanation evaluation metrics was done on the ImageNet validation set and thus took a long time as well, with IAUC and DAUC taking the longest at 1.5 hours per model.

## 4   Results

We chose to classify results that fall within $\pm 0.05$ of the original results as reproducible. Regarding the explanation evaluation metrics, we found that we could not reproduce most results reported in the original paper. The results we acquired do not fully support claim 1. We were able to obtain similar classification accuracy scores on the ImageNet dataset for all models, but we could not reproduce the scores for SCOUTER$_+$ and SCOUTER$_-$ on the Con-text and CUB-200-2011 datasets. Therefore we cannot verify claim 2 with these results. Finally, we were able to reproduce all scores from the confusion matrix metrics on the ACRIMA dataset. While we were able to reproduce the scores, we cannot completely verify claim 3.

### 4.1   Results reproducing original paper

#### 4.1.1   Result 1: reproducing evaluation metric scores

The results of our experiments regarding verifying claim 1 can be seen in Table 3. From this we can see that the area size metric is largely reproducible, but the other metrics are not. Precision deviates not too much from the original scores, but IAUC, DAUC, infidelity and sensitivity differ a lot. Compared to the original scores obtained for the other explanation methods, SCOUTER does not outperform them with our acquired scores. Thus, we were not able to verify claim 1 with our implementation.

#### 4.1.2   Result 2: Reproducing Classification Accuracy

The results of our experiments regarding the verification of claim 2 can be seen in Table 4. As we can see, we were able to reproduce all scores for the models trained on ImageNet. We could also recreate the accuracy scores for the FC model on the other datasets. However, we did not obtain similar scores for any of the SCOUTER models on Con-text

| Model | Area Size | | Precision | | IAUC | | DAUC | | Infidelity | | Sensitivity | |
|---|---|---|---|---|---|---|---|---|---|---|---|---|
| | original | reproduced | original | reproduced | original | reproduced | original | reproduced | original | reproduced | original | reproduced |
| Scouter$_+$($\lambda = 1$) | 0.1561 | 0.1564 | 0.8493 | 0.7898 | 0.7512 | 0.3377 | 0.1753 | 0.4013 | 0.0799 | 0.0006 | 0.0796 | 1.9167 |
| Scouter$_+$($\lambda = 3$) | 0.0723 | 0.1545 | 0.8488 | 0.7949 | 0.7650 | 0.3564 | 0.1423 | 0.4641 | 0.0949 | 0.0001 | 0.0608 | 1.5672 |
| Scouter$_+$($\lambda = 10$) | 0.0476 | 0.1448 | 0.9257 | 0.7870 | 0.7647 | 0.3466 | 0.2713 | 0.4203 | 0.0840 | 0.0601 | 0.1150 | 2.2629 |
| Scouter$_-$($\lambda = 1$) | 0.0643 | 0.0946 | 0.8238 | 0.8481 | 0.7343 | 0.2446 | 0.1969 | 0.4845 | 0.0046 | 0.0012 | 0.0567 | 2.2735 |
| Scouter$_-$($\lambda = 3$) | 0.0545 | 0.0804 | 0.8937 | 0.6686 | 0.6958 | 0.3488 | 0.4286 | 0.3555 | 0.0196 | 0.0961 | 0.1497 | 2.9514 |
| Scouter$_-$($\lambda = 10$) | 0.0217 | 0.0364 | 0.8101 | 0.8968 | 0.6730 | 0.2148 | 0.7333 | 0.4783 | 0.0014 | 0.0028 | 0.1895 | 3.0524 |

Table 3: Explanation evaluation metrics for all different SCOUTER models trained on ImageNet. The original scores are reported in Table 1 in [8]. Scores that diverge more than 0.05 from the original value are highlighted in orange.

and CUB-200-2011. Our trained SCOUTER models perform significantly worse on these datasets compared to what the original paper reported and the scores we obtained for the FC models. Therefore, we did not find full support for claim 2, as we did not find SCOUTER to perform similar to the FC model on Con-text and CUB-200-2011.

| Model | ImageNet | | Con-text | | CUB-200-2011 | |
|---|---|---|---|---|---|---|
| | original | reproduced | original | reproduced | original | reproduced |
| FC | 0.8080 | 0.8086 | 0.6732 | 0.6831 (±0.0156) | 0.7538 | 0.7824 (±0.0274) |
| Scouter$_+$ | 0.7991 | 0.7717 | 0.6870 | 0.5492 (±0.0182) | 0.7362 | 0.4718 (±0.0212) |
| Scouter$_-$ | 0.7946 | 0.7952 | 0.6866 | 0.6093 (±0.0191) | 0.7490 | 0.4143 (±0.0235) |

Table 4: Classification accuracy on various datasets. The original scores are reported in Table 3 in [8] with ResNeSt-26 as backbone. Scores that diverge more than 0.05 from the original value are highlighted in orange. Where applicable, standard deviation is reported in parentheses.

### 4.1.3 Result 3: reproducing ACRIMA confusion matrix evaluations

The results of our experiments regarding claim 3 can be seen in Table 5. We were able to reproduce all results reported in the original paper. However, claim 3 states that SCOUTER achieves a higher score than the FC model in the reported confusion matrix metrics. This was not the case with the results we found. There is a very slight difference between the scores of SCOUTER and the FC model, where in some cases the FC model obtains a marginally higher score than (one of) the SCOUTER models. In the original paper, SCOUTER also only slightly outperformed the FC model. Thus, we have been able to reproduce the reported scores, but these results do not fully support claim 3.

| Model | AUC | | Accuracy | | Precision | | Recall | | F1-Score | | Kappa | |
|---|---|---|---|---|---|---|---|---|---|---|---|---|
| | original | reproduced | original | reproduced | original | reproduced | original | reproduced | original | reproduced | original | reproduced |
| FC | 0.9997 | 0.9993 (±0.0003) | 0.9857 | 0.9843 (±0.0141) | 0.9915 | 0.9897 (±0.0121) | 0.9831 | 0.9811 (±0.0103) | 0.9872 | 0.9836 (±0.0092) | 0.9710 | 0.9561 (±0.0227) |
| Scouter$_+$ | 1.000 | 0.9953 (±0.0034) | 1.000 | 0.9831 (±0.0129) | 1.000 | 0.9718 (±0.0212) | 1.000 | 0.9919 (±0.0096) | 1.000 | 0.9854 (±0.0103) | 1.000 | 0.9566 (±0.0329) |
| Scouter$_-$ | 0.9999 | 0.9989 (±0.0004) | 0.9952 | 0.9856 (±0.0092) | 1.0000 | 0.9896 (±0.0082) | 0.9915 | 0.9768 (±0.0213) | 0.9957 | 0.9876 (±0.106) | 0.9903 | 0.9757 (±0.0208) |

Table 5: Confusion matrix metrics obtained by the models on the ACRIMA datasets. The original scores are reported in Table 4 in [8]. Scores that diverge more than 0.05 from the original value are highlighted in orange. Where applicable, standard deviation is reported in parentheses.

## 4.2 Results beyond original paper

Since the accuracy scores we found for the SCOUTER models trained on CUB-200-2011 and Con-text were so much lower, we wanted to see if training for more epochs would be beneficial. The models did not seem to have converged fully after only 20 epochs. In the examples the authors reported in their code repository, they state 150 epochs for training models on these datasets, so we tested that amount. However, for SCOUTER$_+$ trained on CUB-200-2011 this only resulted in an accuracy of 0.6443, which still deviates significantly from the original score of 0.7362.

## 5 Discussion

Given the results presented above, we did not verify all claims presented in Section 2.
Regarding claim 1, we believe this to be mostly due to the fact that we had to implement most of the metrics ourselves. The authors do not report on how they implemented their metrics and what settings they have used. That means it is highly likely that there exist discrepancies between our code and theirs. It could be the case that they have done some

additional calculations on the metrics, especially sensitivity, since that metric always lies between 0 and 1 in [8], but our found scores do not. Furthermore, the sensitivity scores reported in [16] of which we have used the code also do not necessarily lie in this range. Since our found scores are not in line with what was originally reported, we cannot verify if SCOUTER outperforms other explanation methods.

The fact that we were not able to obtain similar classification scores for both SCOUTER models on Con-text and CUB-200-2011 could be due to the fact that the authors used different hyperparameters than we did. Not all hyperparameters were reported, so in some cases we had to make decisions ourselves. Unfortunately, we did not have the time to run an extensive hyperparameter search. We could thus not verify claim 2.

While we did find similar scores for the confusion matrix metrics on ACRIMA, we could not find support for what claim 3 states: SCOUTER outperforms the FC model on this dataset. The scores we found are very similar between the models, but we would argue that this was the case in the original paper as well. The scores may not fully support the claim that is being made.

Finally, our approach has some shortcomings that is mostly due to time constraints. We did not do three separate runs for training models on ImageNet and thus our findings are based on a single run, which is not ideal. The results on the ImageNet dataset should therefore not be interpreted as final. Furthermore, we were not able to experiment with different backbones and only used ResNeSt-26, which was the main backbone that was used in the original paper.

## 5.1 What was easy

The original paper was well written, making it manageable to understand the SCOUTER model. Moreover, the code for training the models was available. As such, training the models was done with relative ease. A checkpoint system for the models was implemented, meaning training could be stopped and resumed later. The datasets the original authors used are publicly available and straightforward to find and download.

## 5.2 What was difficult

The code of the original authors was devoid of documentation, making it difficult to navigate and pinpoint which part performed what operation. Due to this, we spent a lot of time on any implementation we had to create or extent. Furthermore, the generation of attention maps, arguably one of the most important parts, was hidden somewhere in the code and not documented.

While the code for training the models was accessible, there was no code available for the evaluation metrics. During training, we would encounter a memory allocation error every 8 to 13 epochs, meaning we had to resume from checkpoints. The ImageNet dataset is very large and thus took a lot of time to train. Lastly, there was no code provided for working with the ACRIMA dataset. We had to implement loading the dataset and evaluating the performance ourselves.

## 5.3 Communication with original authors

We sent an e-mail to the authors enquiring about the missing code. However, we did not receive a reply.

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
