# OpenReview forum: "Reproducibility Study - SCOUTER: Slot Attention-based Classifier for Explainable Image Recognition"
_ML_Reproducibility_Challenge/2021/Fall — RC2021_

### Official Review · Reviewer_1A3v · 2022-02-25

**Rating:** 7
**Confidence:** 4

**Review:**

**Scope of reproducibility**

The report defines well the scope of the reproducibility.

**Code**

The authors have used the original code and have added additional metrics and code as required.

**Communication with original authors**

The authors of the report reached out to the original authors but they did not receive an answer.

**Hyperparameter search**

The authors tried to match the original hyperparameters as close as possible, making some decisions on unknown parameters given that they were not described in detail in the original paper and that the original authors did not communicate.

**Discussion on results**

The authors correctly discuss how the results differ from those originally reported and discuss different justifications for these discrepancies.

**Overall rating**

This report is clear and well written. It is a correct reproducibility effort that highlights issues in reproducing the original paper while following a proper methodology. Overall the authors reproduced the experimental setup of the original paper and commented when conditions could have differed and explained why that could be the case. I argue for the acceptance of the report.

---

### Official Review · Reviewer_TzUc · 2022-03-05
**Sheds some light on limitations in the original article, and missing code, but did not manage to fill these gaps.**

**Rating:** 6
**Confidence:** 3

**Review:**

Reproducibility Summary
-----------------------------------
The summary section is complete and quite clear, with the appropriate information, except for the "scope of reproducibility", which simply states the contributions of the original article as 3 main claims, and their goal to "replicate" it.

Minor clarity points:
- l.7, "fully connected model" is confusing, when the model consists of a ResNeSt network, and only the last layer (classifier) is fully-connected.
- l.12, "several datasets", it would have been clearer to state they were the same datasets as the original article, not additional ones.

Scope of reproducibility
--------------------------------
The stated scope is simply to re-run the original experiments. The more extended version in the report clarifies which experiments they mean to reproduce.
The authors mostly adhere to it, to the extent they were able to with the available code.

Code
--------
The authors re-used the original authors repository, except for missing parts:
- the handling of ACRIMA dataset
- the reported metrics, that the authors say they had to re-implement.

This re-implementation:
- re-uses code from other repositories (https://github.com/chihkuanyeh/saliency_evaluation and https://github.com/eclique/RISE), which is mentioned in the text and properly credited in the reproduction repo
- might have some bugs or issues, since the authors report having output values in unexpected ranges

Communication with original authors
-------------------------------------------------
The report mentions trying to get in touch with the original authors, and asking for guidance or code for the missing parts, but not receiving any answer.
There's also an un-answered ticket on the original GitHub (https://github.com/wbw520/scouter/issues/8).

Hyperparameter Search
---------------------------------
Only one hyperparameter has been validated, the number of "slots per class", despite it being provided in the README for most datasets, and it not having a large effect.

The authors did report variance estimates using 3 runs for most experiments, which had not been done in the original article.

Ablation Study
---------------------
No specific ablation study have been performed.

Discussion on results
-----------------------------
The report contains discussion on reproducibility, with:
- the ease of running most training experiments, and
- lack of internal code documentation, and missing code for evaluation and reporting metrics, and for the ACRIMA dataset.

Some results have been confirmed:
- Area size for models trained on ImageNet
- Classification accuracy for ImageNet models, as well as the baseline FC model on ConText and CUB-200
- Confusion metrics on the ACRIMA dataset, although the variance indicates the conclusion is less strong than reported in the original article.

Remaining experiments were deemed "not confirmed", but plausible reasons for that include:
- Not following the hyperparameters in the README.md for ConText and CUB-200 (and not doing a hyper-parameter search either, beyond a small attempt)
- Issues in the implementation or application of the explainability metrics.

These indicate issues in the original paper and code, but I think these could have been better addressed in the report.

Recommendations for reproducibility
--------------------------------------------------
The authors did not explicitly provide recommendations for improving reproducibility.
Implicitly, the need for complete code and author responsiveness are mentioned in the report.

Results beyond the paper
-----------------------------------
No results were provided beyond the paper.

Overall organization and clarity
------------------------------------------
Grammatical issues / organization / proper plots

Conclusion
----------------
This report uncovers several issues in reproducibility of the original work, notably:
- The lack of code for computing and reporting the metrics of interest
- The un-responsiveness of the original authors
- The lack of code and hyper-parameters for the ACRIMA dataset
- The over-optimistic claims of better results on the ACRIMA datasets when no confidence intervals where provided in the original article, and the proposed methods do not seem distinguishable from the baseline.

However, the authors did overlook the instructions and hyper-parameters provided in the README file for the CUB and ConText datasets.
When the reported metrics did not fit the expected range, they did not report attempting to run them on the original baselines (to be able to compare SCOUTER against them), for instance with RISE. There was also no discussion on the potential impact of the model not training well (or the method failing), vs. the metrics not being reported correctly, or attempts to investigate it.

Overall I think the report provides some useful information on the original article, and the problems with it, and corrects it to an extent, but its usefulness is limited by the potential implementation issues and incorrect set of hyperparameter for some experiments.

---

### Meta-Review · Area_Chair_9oxa · 2022-04-09

**Recommendation:** Accept
**Confidence:** 4

**Metareview:**

Reviewers praised this paper's attempts to match the experimental setup from the original work, estimation of variance from multiple runs, and and discussion.

---

### Decision · Program_Chairs · 2022-04-09

**Decision:**

Accept

**Comment:**

Following the recommendation of reviewers and meta-reviewer, the paper is accepted for ML Reproducibility Challenge 2021, and will be published in the upcoming special edition of ReScience Journal.